

# Assessment of the sea surface temperature diurnal cycle in CNRM-CM6-1 based on its 1D coupled configuration

Aurore Voldoire[1], Romain Roehrig[1], Hervé Giordani[1], Robin Waldman[1], Yunyan Zhang[2], Shaocheng Xie[2], Marie-Nöelle Bouin[1,3]

[1]CNRM, University of Toulouse, Météo-France, CNRS, Toulouse, France.
[2]Lawrence Livermore National Laboratory, Livermore, California, USA
[3]Univ. Brest, CNRS, IRD, Ifremer, Laboratoire d'Océanographie Physique et Spatiale (LOPS), IUEM, 29840 Brest, France.

*Correspondence to*: Aurore Voldoire (aurore.voldoire@meteo.fr)

**Abstract.** A single column version of the CNRM-CM6-1 global climate model has been developed to ease development and validation of the boundary layer physics and air-sea coupling in a simplified environment. This framework is then used to assess the ability of the coupled model to represent the sea surface temperature (SST) diurnal cycle. To this aim, the atmospheric-ocean single column model (AOSCM), called CNRM-CM6-1D, is implemented on a case study derived from the Cindy-Dynamo field campaign over the Indian Ocean, where large diurnal SST variabilities have been well documented.

Comparing the AOSCM and its uncoupled components (atmospheric SCM and oceanic SCM, called OSCM) highlights that the impact of coupling in the atmosphere results both from the possibility to take in to account the diurnal variability of SST, not usually available in forcing products, and from the change in mean state SST as simulated by the OSCM, the ocean mean state not being heavily impacted by the coupling. This suggests that coupling feedbacks are more due to advection processes in the 3D model than to the model physics. Additionally, a sub-daily coupling frequency is needed to represent the SST diurnal variability but the choice of the coupling time-step between 15min and 3h does not impact much on the diurnal temperature range simulated. The main drawback of a 3-h coupling being to delay the SST diurnal cycle by 5h in asynchronous coupled models. Overall, the diurnal SST variability is reasonably well represented in CNRM-CM6-1 with a 1h coupling time-step and the upper ocean model resolution of 1m.

This framework is shown to be a very valuable tool to develop and validate the boundary layer physics and the coupling interface. It highlights the interest to develop other atmosphere-ocean coupling case studies.

## 1 Introduction

Because of the many interactions and feedbacks occurring in a general circulation model (GCM), either between parameterizations or between the parameterized subgrid processes and the resolved dynamics, understanding their behaviour or the origin of their systematic errors is often complex. The latter task is even more complicated when GCMs are coupled together (e.g., as in ocean-atmosphere models). In the history of GCM development, simplified versions of GCMs, such as single column model (SCM) consisting of a single grid column of the host GCM, have therefore been widely used (e.g. Betts





and Miller, 1986; Price et al., 1986; Gaspar et al., 1990; Randall et al., 1996; Hourdin et al. 2017; Giordani et al., 2020). Such modelling frameworks ease the development of parameterizations as the resolved dynamics is fully controlled and does not interact with the simulated subgrid processes (Randall et al., 1996; Randall and Cripe, 1999). Besides, SCMs have the great advantage of being computationally very cheap, enabling their use on personal computers and allowing modellers to

rapidly test their ideas and perform numerous sensitivity tests. The reduced set of interactions and feedbacks in SCMs compared to the host GCM, and the possibility to output many diagnostics at model time steps on the model vertical grid also help the modeller to better tackle the cause-and-effect relationship among the parameterized processes and thereby identify model deficiencies. A large number of SCM intercomparisons either for atmospheric SCMs (e.g., Bechtold et al., 1996; Lenderink et al., 2004; Guichard et al., 2004; Cuxart et al., 2006; Klein et al., 2009; Davies et al., 2013; Couvreux et

al., 2015) or oceanic SCMs (Acreman and Jeffery, 2007, Damerell et al., 2020, Reffray et al., 2015) have helped to improve the model parameterizations. This effort has been possible thanks to the availability of high-resolution simulations as reference (e.g., Randall et al., 1996; Couvreux et al., 2021 and reference therein).

SCMs of either the atmosphere or the ocean often require constraints of large-scale circulations and boundary conditions at the surface. Over ocean surfaces, atmospheric SCMs are often used with a prescribed sea-surface temperature

(e.g., Chlond et al., 2004, Neggers et al., 2017) or with prescribed surface fluxes (e.g., Abdel-Lathif et al., 2018). This implies no feedback of the simulated atmosphere on the surface boundary condition as in the real ocean-atmosphere system, and thus limits the potential use of SCMs i22n conditions where the ocean-atmosphere coupling is critical. Similar limitations are also found over land, even though the use of atmospheric SCMs coupled to a land surface model is more common (e.g., Giordani et al., 1996, Bosveld et al., 2014). In the case of oceanic SCMs, the boundary conditions are

similarly provided by prescribing either the atmospheric near-surface parameters or the surface fluxes. Prescribing atmospheric surface parameters induces a strong restoring of the sea surface temperature through the turbulent heat fluxes bulk parameterization and thus constrains strongly the ocean mixed layer heat content. Prescribing surface fluxes appears as a good alternative but surface fluxes are not directly measured and estimations are always uncertain. In addition, prescribed surface fluxes do not account for the thermal adjustment of the atmospheric boundary layer to the sea surface conditions (e.g.

Barnier et al., 1995). To overcome these caveats and properly study the interplay between the atmosphere and ocean boundary layers, only a few studies have so far developed coupled ocean-atmosphere SCMs (Clayson and Chen, 2002; Deppenmeier et al., 2020; Hartung et al., 2018).

The present work seeks to reduce this gap, by developing the atmosphere-ocean SCM (AOSCM) version of the CNRM-CM6-1 climate model (Voldoire et al., 2019). To remain as relevant as possible to its 3D counterpart, both

scientifically and technically, the AOSCM is developed while keeping most of the 3D model technical framework, in particular its coupling interface. Such an AOSCM is also a practicable tool to better understand the coupled ocean-atmosphere feedbacks enabling modellers to disentangle the role of the dynamics from the model physics.

Among ocean-atmosphere coupled processes, the SST diurnal cycle is of special interest and has been the subject of active research over the past decade. Diurnal warm layer anomalies can reach up to 5°C in the Tropics (Ward, 2006; Wick



and Castro, 2020) and have been shown to impact on the atmospheric and oceanic mean state (Itterly et al., 2021; Li et al., 2020) and on climate variability, in particular on the Madden-Julian Oscillation (MJO, Bernie et al., 2008; Seo et al., 2014). Bernie et al. (2005, 2008) highlights a rectification effect of the diurnal SST variability on the mean state: the daily maximum SST is increased as is the upper ocean stratification which further increases the upper layer heat uptake in the

following days leading to a persistent SST warming that can modify the monthly mean SST.

The ability of our GCM to represent this diurnal variability is of interest but difficult to validate in 3D configurations. We therefore take advantage of the AOSCM framework to better understand the ability of the CNRM-CM6-1 atmospheric and ocean vertical physics to represent diurnal oceanic warm layers in the tropics. In this regard, an AOSCM case study based on the Cooperative Indian Ocean Experiment on Intraseasonal Variability in the Year 2011

(CINDY2011)/Dynamics of the MJO (DYNAMO) field campaign (Yoneyama et al., 2013) is developed and serves to highlight the features of the model configuration key to properly capture the SST diurnal cycle. Following previous studies (e.g., Bernie et al., 2008; Ma and Jiang, 2021), the role of the vertical resolution in the upper ocean and of the atmosphere-ocean coupling frequency are emphasized.

Section 2 describes the CNRM-CM6-1 AOSCM, which is referred to as CNRM-CM6-1D hereafter. Based on the

CINDY2011/DYNAMO field campaign and other related data, Section 3 develops the forcing appropriate for an AOSCM case study. It also introduces the data used as reference for analysis. Section 4 discusses the representation of the SST diurnal cycle in either atmospheric or oceanic stand-alone versions of the AOSCM, while section 5 fully makes use of the coupled AOSCM to assess its ability to represent the SST diurnal cycle and how this representation depends on the AOSCM configuration. Section 6 concludes this work.

**2 Model Description**

The objective of this work is to develop a single column version of the climate model CNRM-CM6-1, the CMIP6 version of the CNRM-CM model (Voldoire et al., 2019). This model is composed of ARPEGE 6.3 (Roehrig et al., 2020) for the atmosphere, NEMO3.6 (Madec et al., 2017) for the ocean, SURFEX v8 for the land processes and ocean surface fluxes (Decharme et al., 2019), CTRIP for river routing and GELATO 6 for the sea-ice. These components are coupled through

OASIS3-MCT (Craig et al., 2017). To derive a single column version, we use the SCM version of ARPEGE that has been extensively used to develop and test atmospheric parameterizations (e.g., Abdel-Lathif et al., 2018; Roehrig et al. 2020). The atmospheric SCM uses SURFEX to represent the land surface processes or to calculate the ocean turbulent surface fluxes, depending on the case properties and SCM setup. An SCM version of NEMO has also been used to develop and test oceanic parameterizations (Giordani et al., 2020; Reffray et al., 2015). In the 3D model, the OASIS coupling interface is

implemented within both SURFEX and NEMO (Voldoire et al., 2017). The individual AOSCM components are thus already in place, which eases the development of the fully coupled AOSCM (Fig. 1). Only the CTRIP component, which is fundamentally 2D, is not included in the AOSCM .



In practice, the ARPEGE-SURFEX SCM consists of four identical grid columns, as it allows the SCM to use the same dynamical core as the regional configuration of ARPEGE (e.g., Nabat et al., 2020). Note that this dynamical core

shares the same semi-Lagrangian advection scheme as ARPEGE, while it treats differently the spectral transforms required by the spectral formulation of the model. The vertical advection of the model state variables, when needed, is thus computed as in the 3D model. Similarly the NEMO SCM uses nine identical grid columns. For a given component, the coupling finally considers a single grid cell of it and replicates the associated information to transfer it to all grid cells of the other component. Thus OASIS3-MCT does not perform any horizontal interpolation and only provides communication support to

the coupling. Keeping OASIS3-MCT in CNRM-CM6-1D ensures full consistency between the AOSCM and its 3D counterpart, in particular with respect to the coupling time sequence. Each individual component integrates their own physics in-between two coupling time-steps at which they exchange the relevant coupling fields. The atmosphere SCM thus receives and uses the sea surface temperature as computed by the ocean SCM at the end of the previous coupling time-step, while the ocean SCM receives and uses the surface turbulent fluxes as computed by the atmospheric SCM (SURFEX component) and

accumulated during the previous coupling time-step (asynchronous coupling). The coupling time-step is an OASIS parameter, and in the 3D case, it is fixed to 1 h. In this SCM case study, the impact of choosing a coupling frequency from 5 min to 1 day will be assessed. To be able to disentangle the effect of changing the coupling time-step from changing the model component time-steps, we have fixed the atmospheric and oceanic time-step to 5 min in the SCM, so as to enable the use of a very short coupling time-step. It has however been assessed that changing the component time-steps from their

default values (15 min in the atmosphere and 30 min in the ocean) to 5 min does not alter the SCM results. In principle, the sea-ice model GELATO, which is integrated within NEMO, could be activated in the AOSCM. However, as we focus hereafter on a tropical case study, this has not been tested yet.

In this study, to enable a fair comparison of the Oceanic SCM (OSCM) with the AOSCM, we have run the OSCM jointly with the SURFEX platform so as to ensure a common flux computation using the COARE version 3.0 bulk scheme

(Fairall et al., 2003).

## 3 Development of a CINDY2011/DYNAMO AOSCM case

### 3.1 Selected period and location

The 3-month CINDY2011/DYNAMO field campaign (Yoneyama et al., 2013) was designed to study the Madden-Julian Oscillation initiation in the Indian Ocean, in particular in interaction with the surface ocean. We take this campaign as

an opportunity to develop an AOSCM case permitting to study the representation of the upper-ocean diurnal cycle, which includes frequent and intense diurnal warm layers over the region (e.g., Kawai and Wada, 2007, Matthews et al., 2014). The field campaign provides a wide variety of measurements, including high-frequency soundings from either local islands or



research vessels that were deployed during the campaign, near-surface measurements in the atmosphere and the ocean, and upper-ocean profiles.

130        Ciesielski et al. (2014) derived the various terms of the mass, energy, and water budgets (e.g., vertical velocity, horizontal advections, sub-array diabatic terms) at the scale of two large-scale arrays (~800 km) over the tropical Indian Ocean. This dataset was used in Abdel-Lathif et al. (2018) to force a previous version of the ARPEGE-SURFEX SCM. In contrast, available ocean data mostly comes from the R/V Revelle, which was used as a fixed station at 0°N, 80.5°E for periods of several months. This local data is not sufficient to derive a consistent large-scale forcing of the AOSCM ocean

component. We therefore decided to develop a more local AOSCM forcing, based on the R/V Revelle data. The ship's location at the Equator implies possible strong large-scale advections, which need to be quantified for forcing the AOSCM. This issue is discussed in the next section.

As a result, the AOSCM case developed hereafter focuses on the first 10 days of the R/V Revelle leg 3, i.e. from 13 November 2011, 0000 UTC, to 23 November 2011, 0000 UTC. This period corresponds to a convectively-suppressed MJO

phase followed by the early beginning of a convectively-active phase. A clear motivation for this choice is the occurrence of large SST diurnal cycles during most of this period.

**3.2 Reference datasets**

To assess the model simulations, we use local observations data from the R/V Revelle. SSTs taken are obtained from the ship's intake thermosalinograph at 5 m depth corrected to represent the skin temperature based on the Sea Snake floating

thermistor at 5 cm depth (Edson et al., 2016; de Szoeke et al., 2015). In the following the SST used as reference will be the SST skin temperature.

Reference turbulent fluxes are obtained from the NOAA PSD (Physical Science Division) and provided by C. Fairall and L. Bariteau at hourly frequency. Two products are derived from high-frequency field measurements, either computed using Eddy Covariance (EC) or Inertio-Dissipative (ID) methods. We also use both Conductivity Temperature

Depth (CTD) casts and the Oregon State University Chameleon profiler to get ocean temperature and salinity profiles, as well as Acoustic Doppler Current Profiler (ADCP) records to get current profiles (Moum, 2016). The Oregon State University Chameleon profiler also provides profiles of the Brunt-Vaisala frequency.

Several reanalysis products are also used, both for the atmosphere and the ocean. For the ocean, we have used ORAS5 (Zuo et al., 2019) and Glorys2V4 (Ferry et al., 2012). Both have a resolution of 0.25° with a latitudinal refinement

to 0.15° near the equator. For the atmosphere, we have used the ERA-Interim daily data (Dee et al., 2011).

In this study, as we are more interested by the mean behaviour of the model in simulated SST diurnal cycles, we have chosen to calculate the mean diurnal temperature range (DTR) as the amplitude of the mean SST cycle averaged over the period of analysis. In practice, DTR is the difference between the maximum and the minimum temperature of the mean daily cycle. This choice mainly reduces the values of DTR but does not impact on the main outcomes of this study.



### 3.3 Atmospheric model setup


The large-scale atmospheric forcing of the AOSCM is derived from a constrained variational analysis (CVA), following Zhang and Lin (1997), Zhang et al. (2001) and Xie et al. (2004). The CVA assesses the large-scale mass, energy, and moisture budgets at the scale of a 50-km-radius disk centered on the R/V Revelle, and at the 3-hourly timescale. The 50-km scale is a trade-off between a reduced amount of noise in the forcing and the targeted local scale of the R/V Revelle

measurements used hereafter. The CVA uses the European Centre for Medium Range Weather Forecasts (ECMWF) operational analyses as an input and the surface precipitation measurements retrieved from the Colorado State University's Tropical Ocean Global Atmosphere (TOGA) radar, onboard the R/V Revelle as a constraint. The CVA provides initial profiles of wind horizontal components, temperature and specific humidity, as well as the initial surface pressure, which are used to initialize the AOSCM atmospheric component. It also estimates the horizontal advections of temperature and specific

humidity and the pressure vertical velocity, which are further used as a forcing of the atmospheric column. The SCM zonal and meridional wind components are nudged towards those of the CVA with a 3-hourly nudging timescale. Following Abdel-Lathif et al. (2018), note that the wind, temperature and moisture profiles are extended above 50 hPa using first the ERA-Interim reanalysis (Dee et al. 2011) up to 1 hPa and then the 1976 U.S. standard atmosphere profile (COESA, 1976 - temperature only, wind and specific humidity being set to zero). Above 50 hPa, the horizontal advections and vertical

velocity are set to zero, while temperature and specific humidity are further nudged toward the extended profile. At the surface, the CVA surface pressure is prescribed. In case of atmosphere-only SCM simulations, the R/V Revelle observed SSTs are imposed and thus surface fluxes are computed by the SURFEX bulk parameterization. Table 1 summarizes this model physical configuration.

SCM simulations may be strongly constrained by the prescribed forcing. However the latter often has large

uncertainties because of the very few in-situ observations, which weakly constrain the input data of the CVA (e.g., the ECMWF analyses) or which more directly enter the CVA algorithm. This may sometimes question the detailed comparison between SCM simulation output and other, independent data. As an attempt to address the forcing uncertainties, 10-member ensembles of SCM simulations are performed by introducing a weak and random noise in the forcing large-scale advection and vertical velocity fields. This should avoid focusing the upcoming analysis on small, most likely insignificant, differences

between simulations or between simulations and reference datasets. Note that all experiments discussed hereafter have been listed in table 2.

### 3.4 Ocean model setup

To design the ocean column forcing, we use the NEMO single column version forced by near-surface atmospheric measurements collected at the R/V Revelle at a 10-minute frequency, namely air temperature, specific humidity, wind speed,

surface pressure, surface precipitation, and surface downwelling longwave and shortwave radiation. The atmospheric forcing is thus representative of the local R/V Revelle scale, which the OSCM simulations are intended to match.





The ocean column state is initialized using in-situ CTD data for temperature and salinity and from ADCP measurements for currents. As these observations extend to 250m and 150m depth, respectively, the ORAS5 reanalysis data is used below these levels. No spurious gradients are generated near the merging depth, as ORAS5 is found to simulate profiles very close to those observed at the R/V Revelle. Moreover, we are mainly interested in the near surface layers, and these are weakly impacted by the oceanic state below 150m over the simulated 10 days (not shown).


As a first guess, the ocean SCM is run without imposing any large-scale advection (horizontal nor vertical) using the default CNRM-CM6-1 NEMO configuration (Voldoire et al., 2019). The simulated daily-mean SST time series warms during the first three days by 0.8 °C, and then approximately follows the observed SST time evolution (Figure 2a, purple versus black lines). The surface warm bias extends up to 10 m while a cold bias develops below 30 m (Figure 3a), thus suggesting a lack of downward heat transfer. The mean salinity (Figure 3b) is accurate up to a few meters. Below about 5 m, the model has a fresh bias of 0.1 psu with respect to the CTD observations, which is however within the range of uncertainty as given by the two ocean reanalyses. The Chameleon data collected at the RV/Revelle provides an estimate of the Brunt-Vaïsala frequency vertical profile (Figure 3c), which is well captured by the OSCM between 100 m and 10 m. Above, the model clearly overestimates the stability.



Several deficiencies may explain the OSCM warm bias in the upper ocean: deficiencies in the OSCM forcing setup such as a missing large-scale advection term or underestimated current that does not generate enough turbulence, deficiencies in the OSCM physics, such as a flaw in the vertical mixing parameterizations (Moulin et al., 2018) or incorrect solar radiation penetration in the ocean.

In this region, McPhaden and Foltz (2013) show that the importance of the horizontal advection of heat could depend on the year considered. As the case-study is on the Equator, surface currents are not negligible and horizontal heat advection may play a crucial role here. Estimates of the horizontal heat advection are computed, based on either the Glorys or the ORAS5 ocean reanalysis. Although the surface current is intense near the R/V Revelle location, the heat advection remains weak and its sign has a large spatial and temporal variability (not shown). Thus it does not provide any systematic heating or cooling of the upper ocean along the studied period. It is also weakly consistent in time and space between the two reanalyses. Note that it is even more difficult to get a reliable estimate of the vertical heat advection.



Given the large uncertainty of heat advection estimates, an idealized framework is set up to assess the heat advection potential impact. Several sensitivity experiments are performed with heat advection profiles constant in time and along the vertical. They indicate that a heat advection of -0,1°C/day is needed to cancel the warming drift (simulation "Ocean vadv" on figure 2a). The cooling extends to 10 m depth but the upper-ocean stability remains overestimated (Figure 3c) which suggests a lack of parameterized vertical mixing.


In this single column configuration, after initialisation, currents are only sustained by wind stress surface fluxes. We may hypothesize that currents are weaker than in 3D configurations and that this source of turbulence is under-represented in this 1D configuration. It has been tested to impose the initial current profile all along the simulation but this does not impact



much the thermal profile nor the upper-ocean stability (not shown). In practice, currents mainly generate turbulence through current shear which is negligible in the first 10 meters of the ocean here.

The vertical eddy diffusivity turns out to be always set to its background value of 1.2 10-5 m2 s-1. An increase of this background vertical eddy diffusivity by a factor 10 (i.e. to 1.2 10-4 m2 s-1, "Ocean tuned" experiment on Figures 2 and 3) leads to a mean vertical profile of the Brunt-Vaïsala frequency much closer to the Chameleon estimates (Figure 3c). The

surface temperature evolution is also improved, with a warm bias reduced from 0.6°C to 0.3°C on average (Fig. 2a). The impact on the mean vertical profile of temperature is rather weak. Indeed, the change of Brunt-Vaïsala frequency profile mainly reflects a change in the diurnal cycle evolution of processes. This Brunt-Vaïsala frequency change reflects an increase only at night when the stratification can be eroded whereas during the day, the upper ocean remains very stable due to the large incoming solar radiation (not shown). The nighttime reduced stability favours mixing and thus night time

cooling. This results in a large decrease of the mean DTR (Figure 2b) which is overestimated in the CM6 experiment (1.5°C) and which is reduced to 1.0°C in the "Ocean tuned" experiment, closer to observed estimates of 1.1°C on average over the 10-day period.

As our objective is clearly to focus on the ocean-atmosphere coupling, we have not investigated further the ocean mixing process representation in the model. However, the tests shown here clearly demonstrate the need to keep on working

on ocean mixing processes and that the OSCM framework is relevant to test new parameterizations. Observations sampling the diurnal cycle in the sub-surface ocean are also critically needed. Additionally, the proper set-up of such ocean configurations requires reliable but challenging estimations of the large-scale advection forcing of the oceanic column. In the remaining of this study, the ocean is set to the "Ocean tuned" configuration where the background eddy diffusivity is increased by a factor 10.

**4 SST diurnal cycle in uncoupled configurations**

This section focuses on the representation of the SST diurnal cycle in uncoupled configurations of both the ocean and the atmospheric model. The aim is to discuss the key elements necessary to realistically represent the SST diurnal cycle in the respective components. As a base for this analysis, we first discuss the observed SST and surface turbulent fluxes observed during the period of our case study.

**4.1 Observations**

During the studied period, we observe large diurnal SST amplitudes of more than 1.5°C over the first 4 days, then their amplitude decreases probably due to the presence of convection and precipitation the last 5 days (Fig. 4). Averaged over the 10 days period, the mean diurnal cycle of SST as an amplitude of 1.1°C. Note that the amplitude of the mean SST diurnal cycle is not equivalent to the mean amplitude of the SST diurnal cycle as it is a non-linear calculation and as its

amplitude varies along the 10 days considered.



The surface flux estimations based on either the EC or ID methods are rather noisy and it is difficult to detect a diurnal cycle in the raw time-series, even averaged over the 10 days, the mean diurnal cycle is rather noisy (Figure 6b-c). The standard deviation of the mean diurnal cycle is 43 W.m-2 on average for the latent heat flux and 10 W.m-2 for the sensible heat flux, a similar amplitude as the respective mean diurnal range (66 W.m-2 for the latent heat flux and 14 W.m-2

for the sensible heat flux). Indeed, Marion (2014) raises that these flux estimates are not accurate under the weak wind conditions of this period. Nevertheless, this dataset provides an idea of the amplitude and phasing of the diurnal turbulent flux cycle.

### 4.2 In the atmosphere

In an atmospheric model, the SST is a forcing, thus the representation of the SST diurnal cycle is generally

conditioned by the frequency of the product used as a forcing. Here in the reference atmospheric experiment (Atm SSTObs 1h), the forcing is taken from hourly SST observations at the R/V Revelle (Fig 4a). In this experiment, the model simulates episodes of precipitation from the 18th of November as in observations, albeit with an underestimated amplitude. The mean latent heat flux (70 W.m-2) and the mean sensible heat flux (10 W.m-2) are also weaker than observed estimates (88 W.m-2 and 11 W.m-2 respectively). Event if observed values are rather uncertain, this underestimation can also be attributed to an

underestimation of the mean wind amplitude simulated (1.5 m s-1 with a standard deviation of 0.7 m s-1), much weaker than the observed mean which reaches 2.1 m s-1with a standard deviation of 1,2 m s-1. As the atmospheric winds are nudged towards the CVA, this probably highlights that the atmospheric forcing is representative of a larger region (50 km) than the local fluxes measurements. This implies that atmospheric simulations are not fully comparable to local observations given that they are representative of a different spatial scale. To take this different scale into account, mean vertical profile of

temperature and relative humidity are compared both to the Era-Interim reanalysis (Fig. 5, Dee et al., 2011)) and to the local radio-sounding measurements. It shows that the model is biased cold compared both the reanalysis and local observations, whereas the relative humidity bias is relatively weak and comparable to the difference between the local soundings and the reanalysis except near the surface where the model is significantly moister than the two references.

To assess the effect of forcing the diurnal SST cycle, we have performed a companion experiment using daily

averaged observed SST (Atm SSTObs 1d). The impact of cancelling the SST diurnal cycle on the mean surface heat flux is very weak: the sensible heat flux is not impacted, the upward long-wave heat flux is only slightly decreased by less than 0.1 W m-2 and the latent heat flux is decreased by 0.5 W m-2 (not shown). Similarly, the simulated impact on precipitation is weak, within the ensemble range (Fig. 4b); the mean impact on the atmospheric mean temperature and relative humidity profiles (Fig. 5) is also negligible. Even if the mean state is not changed, the inclusion of a SST diurnal cycle may impact the

diurnal cycle of the surface heat fluxes and of the atmospheric profiles. We mainly observe an impact on the turbulent heat flux diurnal cycle: the latent heat flux diurnal cycle decreases (in absolute value) by more than 16 W m-2 whereas the sensible heat flux amplitude is only reduced by 1 W m-2 . The latent heat flux daily maximum is better phased when forcing with hourly SSTs. The impact on the tropospheric temperature and relative humidity mean diurnal cycles are shown on



figure 7. The surface temperature cooling at night and warming at daytime is limited to near surface layers below 925hPa.

Similarly, we observe a near surface drying at day time and a moistening at night time. There is no clear significant impact above 925hPa. In this 1D setup, the SST diurnal cycle does not imprint on the mid-troposphere nor on the precipitation. It is likely that the large-scale forcing largely limits any feedback from the surface in this 1D configuration.

**4.3 In the ocean**

In contrast with the atmosphere-only configuration, in the ocean-only configuration the SST is prognostic. The SST

evolves according to the heat budget of the oceanic top layer, it is thus representative of the first layer. It mean that modelled SST represents 1-m depth averaged temperature in the reference configuration. We may wonder if the simulated SST would better match the observed skin SST if the model resolution was increased near the surface. Besides, the role of vertical discretization has been raised in many studies (Bernie et al., 2005; Hsu et al., 2019; Ge et al., 2017) but these studies generally test resolution between 1 m and 10 m. Hsu et al. (2019) assess the diurnal SST representation in the ACCESS-S1

model which also uses NEMO at a 1-m vertical resolution. They suggest that flaws in representing the diurnal warming may be due to insufficient vertical resolution or to deficiencies in vertical mixing in the NEMO model.

The column model allows us to tackle this question relatively easily. We thus extend the analysis already made in Bernie et al. (2005) to higher vertical resolution. The way the vertical coordinate is defined in CNRM-CM6-1 (ln zco case in Madec et al., 2017) makes it complicated to change the ocean vertical resolution (as also raised in Hsu et al., 2019). To ease

the change in vertical grid scale, we have moved to a uniform grid representation and limited the depth to 135 m. To check the effect of changing the vertical grid formulation and reducing the ocean depth, we first perform an experiment in which the vertical resolution is set uniformly to 1 m (Ocean v1m) as in the first layer of the reference experiment. Then we have increased the vertical resolution to 1 cm (Ocean v1cm) and reduced it to 10 m (Ocean v10m).

The results from these simulations are compared to the reference experiment on figure 8. We verified that the

uniform vertical discretisation with 1 m resolution behaves similarly as in the control simulation that used a more complex discretisation but a similar resolution near the surface. As already shown in former studies, a coarser vertical resolution of 10 m strongly reduces the DTR to 0.3°C. With a 1-m resolution, the mean DTR is 1.0°C, only slightly less than the 1.1°C obtained with 1-cm resolution, which is close to the observed estimate. Note also that increasing the resolution has a small impact on the timing of the maximum which is advanced by 1 hour.

To get a more quantitative assessment of the DTR representation depending on vertical resolution, figure 9 provides the percentage of the observed DTR amplitude simulated by the OSCM for a large set of vertical resolution values (blue dots, between 1 cm and 10 m). Between 1m and 10m, this study confirms Bernie et al. (2005) findings (their Fig. 10) with a rapid decrease of simulated SST amplitude with decreasing resolution. With resolutions coarser than 4m, the ratio of SST amplitude drops below 50%. In contrast, when resolution increases to thinner values than 1 m the increase in DTR is rather

small. The grey line represents the DTR obtained in the 1cm resolution simulation but averaging the temperature over the corresponding depth. The amplitude of the DTR representation corresponds well to that of the corresponding resolution





experiment meaning that when running coarse vertical grid resolution, the lowest DTR obtained is due to the fact that it represents a larger "bulk" and not by a flaw of the existing subgrid ocean processes representation. If we compare the DTR obtained at 5 m depth in the 1 cm resolution experiment with the DTR measured by the thermo-salinograph at the same depth, the amplitude of the DTR simulated is about half of the observed DTR (not shown). This suggests that the diurnal processes are concentrated in a shallower layer than in observations. This also tends to show that improving the representation of the nighttime cooling necessitates to better represent the processes involved (Moulin et al., 2018).

This sensitivity test shows that using a 1-m vertical resolution in an ocean model is a good compromise in state-of-the-art models. Even with non-uniform vertical coordinates discretization, reaching a 1cm resolution near the surface would be numerically unfeasible in global 3D climate models. In such models, the near-surface effect could be well introduced using or adapting warm-layer parameterizations (Zeng and Beljaars, 2005; Bellenger et al., 2017; Gentemann et al., 2009; Scanlon et al., 2013) as proposed in Yang et al., (2017).

## 5 SST diurnal cycle in coupled configuration

### 5.1 Impact of the coupling

The coupled reference experiment is based on the stand-alone configurations discussed in the previous section with the "Ocean tuned" set-up for the ocean component. In the reference coupled experiment, the coupling time step is set to 5 min so as to compare the effect of the coupling in the coupled configuration where the asynchronicity is the lowest. The effect of increasing the coupling time-step will be discussed in the next section.

Figure 10a shows the SST evolution for the coupled reference experiment along with reference uncoupled experiments and R/V Revelle observed SSTs. The coupled model SST shows a clear diurnal cycle and follows closely the behaviour of the uncoupled ocean experiment (Ocean Tuned) with a similar warming trend. The simulated SST has a warm bias (+0,2°C), of similar amplitude to that in the ocean forced case (+0,2°C). The warming trend probably results from a lack of nighttime mixing in the case of strong upper ocean stratification as shown in Brilouet et al. (2021).

In the model, the intensity of the diurnal cycle does not change much during the period unlike in observations. The model simulates precipitation events (Fig. 10b), albeit of smaller amplitude, but they do not seem to be associated to a reduction in DTR as in observations. There are several reasons that could explain this weak impact: the simulated precipitation events may be too weak to impact the surface turbulent fluxes that could alter the SST diurnal cycle; or missing processes in the model that should impact the surface turbulent fluxes. For instance, convective precipitation is usually associated to gusts that are known to increase the turbulent fluxes (Godfrey and Beljaars, 1991); such a gustiness effect is not included in the CNRM-CM6-1 model and may explains this weak impact of precipitation events on DTR.

The simulated precipitation is similar between the coupled and the atmosphere-only configurations (Fig 10b). The SST diurnal cycle shape (Fig. 11a) also looks like the one of the ocean-only simulation, the mean DTR being only slightly





reduced to 0.9°C. The latent heat flux mean daily cycle is very similar to the one simulated by the atmosphere-only simulation (Fig.11b). The most striking feature is the large discrepancy between the mean latent heat flux estimated in the

ocean forced simulations compared to the atmospheric and coupled simulations. This difference is well explained by the surface ocean forcing used: the ocean-only experiment is forced by the R/V Revelle's local observations whereas the coupled experiment interactively computes variables and fluxes at the air-sea interface. In particular, we have shown in section 4.2 that the wind forcing explains such a difference.  In the ocean, the difference in turbulent heat flux between the forced and coupled experiment does not impact much on the simulated SST, it reflects well that the ocean vertical profiles of

temperature and salinity are only weakly impacted by the coupling and the change in surface turbulent fluxes.

Figure 12a shows the evolution of the difference in daily mean atmospheric vertical profile of temperature between the coupled experiment and the atmosphere forced experiment. After the first 2 days, a warm anomaly develops near the surface and progressively extends up to the lower free troposphere by the end of the 10-day period. This pictures the impact of the surface warm anomaly on the atmospheric column. There are much less significant impacts on the relative humidity

daily mean profile (Fig 12c). Having removed the mean biases, the impact of the coupling on the atmospheric diurnal cycle is weak and non-significant. To summarize, the first effect of the coupling is a change in the mean state, with a very weak impact on the simulated diurnal cycle. The coupled simulation surface temperature follows the ocean forced experiment temperature evolution, in contrast to the surface turbulent heat fluxes, which mainly follow the atmospheric forced simulation. The latter are more impacted by the change in atmospheric surface wind than by the change in SST.

### 5.2 Impact of the coupling frequency

Figure 11a emphasizes that in the coupled experiment, the SST daily peak occurs between 12h LT and 15h LT relatively in phase with observations, even if there is probably a short delay of less than 1 hour. In the reference coupled experiment, the coupling time-step is 5 min, which is much less than in state-of-the-art global climate models. In 3D climate models, the individual component time-steps are larger than in this single AOSCM configuration and do not allow us to

couple at such high frequency. Hsu et al. (2019) show that with a 1-h coupling, they still have deficiencies in the phasing of the SST maximum but could not go far beyond in their AOGCM. Here the AOSCM configuration allows us to tackle this question with more details. Increasing the coupling frequency from one day to higher frequencies has been done in several CMIP6 climate models but with the use of a 3-hour coupling time-step (Li et al., 2020; Sellar et al., 2020) or a 1-hour coupling time-step (Danabasoglu et al., 2020; Mauritsen et al., 2019). Tian et al. (2019) and Li et al. (2020) both report an

improvement on ENSO representation, especially when switching from a daily to higher coupling frequency. The effect of using a 3-h coupling time-step over a 1-h coupling time-step is however not discussed. The 1D coupled configuration is a relevant tool to highlight the first-order consequences of such a choice. Here, we run a set of experiments in which we only change the coupling frequency, exploring 5 min, 15 min, 1 h, 3 h and daily coupling time-steps.

Figure 13 shows the SST mean daily cycle for these sensitivity experiments. With a daily coupling, unsurprisingly

the SST is held constant all the day to 30.3°C. In all other experiments with an infra-daily frequency the mean SST is similar





(30.4°C). This means that the rectification effect raised in Bernie et al. (2005) may be present but limited to 0.1°C. For coupling time-step between 5 min and 1 h, the DTR is similar (0.9°C) and drops relatively weakly with the 3 h coupling to 0.8°C. The main difference between the experiments with an intra-daily frequency comes from the time of maximum SST, which is delayed consistently with the increased coupling period. Indeed, with the asynchronous coupling algorithm used in
the model, the ocean model uses the solar heat flux calculated in the atmospheric model over the former coupling time-step. Thus increasing the coupling time-step increases the delay by which the ocean model sees the solar radiation diurnal cycle. The daily maximum is reached around 14h LT for the 5-min coupling time-step, in relative agreement with in situ observations. With a 15-min coupling time-step, the maximum is only marginally delayed. With a 1-hour coupling time-step, there is a delay of 2 hours. With a 3-hour coupling time-step the delay is increased to 6 hours.

In our coupled model, the rectification effect is relatively weak compared to the estimation of 0.34°C given in Bernie et al., (2005); however, there are several differences in our study that may explain such a difference in amplitude. We may wonder if the coupling could reduce this rectification effect or if such a rectification effect is less active in the present case study due to different processes at play. To disentangle the coupling effect, we perform additional ocean forced experiments driven by the atmospheric state obtained in the 1h coupled experiment (Table 2). A first experiment uses the
atmospheric forcing at a 1-h time-step (Ocean 1h-forcing) and a second one uses the same forcing but after being daily averaged (Ocean daily avg forcing). Figure 14 shows the evolution of the SST and its daily mean in the coupled experiments and in the ocean forced experiments. First, the difference between the 1 h and daily coupled experiments (Fig 14a) is not increasing all along the period as one would expect if the rectification effect was at play. The daily coupled experiment is colder the first two days, then reaches the mean temperature of the coupled 1 h experiment and even get warmer the
following 5 days. The first day colder SST in the 1-day coupled experiment compared to the 1-h coupled experiment seems attributable to a longer time needed to adapt to the atmospheric warming due to the "weak coupling" induced by the fact that the ocean receives the new atmospheric state with a one-day delay. It does not picture a clear rectification effet.  In the ocean forced experiment, the behaviour is closer to what was shown in Bernie et al., (2005), with the daily forcing experiment being colder by 0,2°C than the hourly forced experiment the first 9 days of the period. Note also that the difference reverses
quickly when the amplitude of the diurnal cycle weakens at the end of the period. This tends to show that the rectification effect is weaker than expected in our case study but more importantly, this effect is not clear anymore in coupled mode.

Figure 15a-b shows the impact of introducing the SST diurnal cycle in the atmospheric column in coupled mode so as to compare with the results in atmospheric forced mode (Fig. 7). The effect of high frequency coupling compared to daily coupling is qualitatively similar with impacts limited to the near-surface atmospheric layers. The intensity of the signal is
weaker consistently with the underestimated DTR in the coupled simulation. We also observe an impact on the diurnal evolution of oceanic temperature in near surface layers (Fig 15c). The daily warming of the ocean upper layers starts from 10h LT several hours earlier than in the first atmospheric layers. This pictures well that the ocean surface temperature warms due to increasing solar radiative heat flux and that the surface temperature warming then feeds back on the atmosphere afterwards.





Figure 16 illustrates the impact of changing the coupling frequency from 5 min to 1 hour (resp. 3 hour) on ocean and atmospheric columns. This can be viewed as the error made when the coupling time-step is fixed to 1 hour (resp. 3 hour) as in 3D models, since with a 5-min time-step we have shown that the SST diurnal cycle matches well the observed one. As expected, the impacts are larger for the 3-hour frequency than for the 1-hour frequency. In both cases, they are limited to the lower troposphere. The impact can be regarded as a delay of the diurnal cycle, with reduced ocean temperature between 8h LT and 15h LT and increased the rest of the day.

We could expect that the phase change of the daily cycle on the atmospheric temperature could impact the cloud cover daily cycle but it is not the case in our experiment (consistently with the weak impact seen on relative humidity).

To summarize, from the ocean point of view, the coupling as a very weak impact on the simulated SSTs. The rectification effect shown in Bernie et al., (2005), which is present in ocean forced mode disappears when coupling with the atmospheric model. From the atmospheric point of view, the effect of the coupling can be decomposed into two effects, first the coupling leads to a mean state change, the SST in coupled mode being biased as in the ocean forced experiment. The mean state change is relatively similar for all coupling time steps. The second effect is linked to the sub-daily SST variability in itself. This second effect has been assessed both in atmospheric forced mode (Fig. 7) and in ocean-atmosphere coupled mode (Fig. 15). The effect of the SST diurnal cycle is very similar and limited to the atmospheric boundary layer below 900 hPa in both configurations on the atmospheric temperature and relative humidity, the signal being weaker in coupled mode.

**6 Discussion and conclusion**

To ease the CNRM-CM model development, we derive from the full 3D coupled GCM an atmosphere-ocean single column version called CNRM-CM6-1D. This configuration consists in coupling the already existing column versions of the atmospheric and ocean models used in CNRM-CM. The 1D model coupling follows closely the global model coupling setup, enabling the study of the coupling between the ocean and the atmosphere as in the 3D model but in a more constrained framework. Running such a model can be done on a common personal computer in a few minutes. Thus it is possible to perform numerous tests that ease debugging and allows one to run many sensitivity tests. Such a configuration is fully relevant to investigate in detail some of the feedbacks between individual components or between parameterizations. When developing new parameterizations, it is also much easier to implement and test them in this 1D configuration.

As a first step, we illustrate here the use of this model to discuss the necessary elements to be included in 3D climate models to properly represent the sea surface temperature diurnal cycle. To this aim, the 1D configuration has been implemented for a CINDY2011/DYNAMO case study. This case study is particularly relevant since we observe large SST diurnal variations during the period. A specific large-scale atmospheric forcing representative of a 50-km radius around the R/V Revelle is used to focus on the local scale.





For the ocean, there is no specific large-scale forcing available and it is not straightforward to derive such a forcing. In its standard configuration, the model overestimates the upper ocean stability and the surface warming. This strong stability is not reduced by imposing a negative trend in temperature mimicking a missing large-scale advective forcing. We therefore increase the background eddy diffusivity to artificially enhance the unresolved turbulent mixing. This results in a
better match with the observed ocean stability profiles and a more realistic DTR. We do not conclude from this that the background eddy diffusivity should be changed in the 3D model, as a deeper analysis over a wider variety of cases is required. In particular, the wind forcing observed at the R/V Revelle during the CINDY2011/DYNAMO campaign is relatively weak and does not span the variability observed nor simulated in the 3D model. This result points out, however, to the need for an improved representation of near-surface mixing processes in ocean models, especially under stable
conditions. This highlights that such a 1D configuration is a relevant tool to investigate the parameterization of ocean mixing processes.

In this case study, it has been possible to run the model for periods longer than 10 days without imposing a large-scale ocean circulation. However, it should be highlighted that this is probably not the case everywhere in the ocean and thus the use of this model for other case studies would require to get such a large-scale ocean forcing. It should be stressed,
though, that the neglect of large-scale circulation is a common practice for single-column ocean modelling experiments (e.g., Reffray et al., 2015 and references therein).

The case study used here is particularly relevant to study processes related to the SST diurnal cycle. In an atmospheric forced model, the SST is a forcing and the capability of representing its diurnal cycle mainly depends on the availability of such datasets. The impacts of including the SST diurnal cycle are relatively weak, but could be larger in a less
constrained modelling framework. To take them into account, it could be valuable to implement a parameterization of the diurnal warm layer (Gentemann et al., 2009; Scanlon et al., 2013; Bellenger et al., 2017; Yang et al., 2017). Such an implementation could be easily validated based on this 1D case study. Such a parameterization could also be used in coupled configurations: with 1-m ocean vertical resolution the simulated DTR captures 90% of the observed estimate, and therefore a warm-layer parameterization may help improve the DTR amplitude.

The pure effect of representing the SST diurnal cycle in atmospheric forced and coupled simulations is similar and limited to the near surface layers both in the atmosphere and in the ocean. We could have expected to find an impact of the SST diurnal cycle on convection as found in Zhao and Nasuno (2020), but it was not the case here. We may hypothesize that, in the present 1D setup, the convection is mostly constrained by the large-scale forcing and that surface temperature would impact convection through changes in vertical motion that is not interactive here.

In our 1D configuration, the impact of coupling on the ocean is very weak, as pictured by the similar evolution of SSTs. Additionally, the "rectification" effect highlighted in Bernie et al., (2005) due to the representation of the diurnal temperature evolution is not present in the coupled experiments. As a result, in the atmosphere, the impact of coupling is nearly directly the sum of the impact of changing the mean state in SST as in the ocean forced simulation plus the effect of introducing the diurnal SST cycle, without any coupled feedbacks. This is probably the absence of dynamical feedbacks in



1D configurations. Such a configuration is of great value to assess new developments and disentangle their first order impacts but it remains a simplified approach to be complemented by 3D studies.

Generally, in 3D simulations, atmosphere-only simulations are done using low-frequency varying SSTs and thus when we compare an atmosphere forced experiment to a coupled experiment, both the effect of introducing a infra-daily varying SST and the pure coupling effect are introduced without being clearly assessed separately. In the 1D column
configuration it is easy to disentangle these two effects, and this should be better highlighted in studies with full GCMs.

In 3D models, reducing the coupling time-step can be expensive in terms of computational cost and it is generally set to 1 hour or 3 hours. Thanks to the 1D configuration, we have been able to assess the effect of such choices. The effect of changing the coupling period is to delay the timing of the daily maximum SST without impacting the DTR. We have shown that with a 5-min or 15-min coupling time-step the models simulate well the timing of the daily maximum SST. With a 1-
hour coupling time-step the delay is limited to 2 hours, with a 3-hour coupling time-step, the delay extends to 5 hours which becomes important relative to the day length.

Such 1D coupled configurations are particularly relevant to perform studies related to the time coupling schemes. It is a very practical tool to test new approaches like Schwartz methods that enable correcting the mismatch in between components due to asynchronous coupling through an iterative method (Marti et al., 2020) and to assess the effect of
different coupling algorithms from purely sequential to asynchronous. It is also an interesting tool to assess the effect of surface flux parameterizations on both ocean and atmospheric boundary layers.

More generally, to be useful for the development and evaluation of surface flux and boundary layer parameterizations, and more generally for modelling choices related to the vertical physics, there is a need to get several coupled 1D case studies as it is done for the atmosphere 1D case studies. Following an initiative from the DEPHY
programme (http://www.umr-cnrm.fr/dephy/), the community is currently developing common standards for ASCM input forcing, in order to facilitate sharing and implementation of the wide library of currently-available ASCM cases. A similar effort would clearly benefit future AOSCM case studies.

**7 Code availability**

ARPEGE-Climat is only available to registered users for research purposes only. SURFEX v8 is distributed using a
CECILL-C Licence (http://www.umr-cnrm.fr/surfex). OASIS3-MCT can be downloaded at https://oasis.cerfacs.fr/en/. NEMO v3.6 can be downloaded at http://www.nemo-ocean.eu/ after a user registration on the NEMO website. Due to the restricted availability of ARPEGE-Climat, CNRM-CM6-1D is only available to registered user on demand to the corresponding author.



## 8 Data availability

Data from the CINDY2011.DYNAMO campaign were downloaded from https://data.eol.ucar.edu/.
Outputs from numerical experiments analyzed are available from https://doi.org/10.5281/zenodo.5772667.

## 9 Author contribution

    AV developed the CNRM-CM6-1D model, made the analysis and wrote the paper, with inputs from RR and RW.
RR developed the atmospheric set-up to force the atmospheric column model. HG and RW brought their expertise in ocean
processes. YZ and SX developed the atmospheric forcing. MNB gathered the atmospheric and turbulent fluxes data and
provided expertise on their use.

## 10 Competing interests

The authors declare that they have no conflict of interest

## 11 Acknowledgements

This research has been supported by the ANR project COCOA (grant no. ANR-16-CE01-0007TS5). R. Roehrig
acknowledges support by the GdR DEPHY. S. Xie and Y. Zhang were supported by ARM program and the Atmospheric
Systems Research (ASR) program in the Office of Biological and Environmental Research, Office of Science, US DOE.
Lawrence Livermore National Laboratory is operated for the US DOE by Lawrence Livermore National Security, LLC,
under Contract DE-AC52-07NA27344.

This study has been conducted using data from the U.S. Department of Energy (DOE) Atmospheric Radiation
Measurement (ARM) program and E.U. Copernicus Marine Service Information (CMEMS). Data from the
CINDY2011.DYNAMO campaign were provided by NCAR/EOL under the sponsorship of the National Science Foundation
(https://data.eol.ucar.edu/). We acknowledge the kind help of C. Fairall and L. Bariteau in using these data.



**Tables**

| Component | Physical parameter | Configuration |
|---|---|---|
| Atmosphere (ARPEGE) | Horizontal momentum | Ekman dynamics (Coriolis, vertical physics : turbulence, convection) restored toward a prescribed wind profile |
|  | Dry static energy & humidity | - Vertical physics: turbulence, convection, radiative<br>- Forcing by nudging the upper tropo (above 50hPa); prescribed horizontal advection trend; prescribed vertical velocity profile |
| Air-sea interface (SURFEX) | Horizontal momentum | Turbulent Bulk formulation |
|  | Water | - Precipitation: from atmosphere<br>- Evaporation: turbulent Bulk formulation |
|  | Heat | - Radiative: from atmosphere and ocean<br>- Sensible and latent: turbulent Bulk formulation |
| Ocean (NEMO) | Horizontal momentum | Ekman dynamics (Coriolis, vertical physics : turbulence, convection) |
|  | Temperature & salinity | Vertical physics: turbulence, convection, internal waves, radiative (only for temperature) |

**Table 1: Physical configuration of CNRM-CM6-1D.**

| Configuration | Simulation name | Details |
|---|---|---|
| Ocean only | Ocean CM6 | CNRM-CM6-1 ocean physics and parameters |
| | Ocean Tuned | AS Ocean CM6 with increased background eddy diffusivity (x10) |
| | Ocean Vadv | As Ocean CM6 with imposed vertical heat advection 0.1°C/day |
| | Ocean v1m | As Ocean CM6 with uniform vertical level resolution set to 1m and depth limited to 135m |
| | Ocean v10cm | As Ocean v1m with resolution 10cm |
| | Ocean v1cm | As Ocean v1m with resolution 1cm |
| | Ocean 1-h forcing | As Ocean tuned but forced by the atmospheric state obtained in the coupled experiment Atm SSTObs 1h |
| | Ocean daily avg forcing | As Ocean 1-h forcing but with the forcing daily averaged |
| Atmosphere only | Atm SSTObs 1d | Atmopheric experiment forced with R/V Revelle observed daily mean SST |
| | Atm SSTObs 1h | As former using R/V Revelle observed SST (1h) |
| Coupled | Coupled | Reference coupled experiment, with 1-h coupling time-step |
| | Coupled 5min | As coupled with a 5-min coupling time- step |
| | Coupled 15min | As coupled with a 15-min coupling time-step |
| | Coupled 3h | As coupled with a 3-h coupling time-step |
| | Coupled 1day | As coupled with a daily coupling time-step |

**Table 2: List of experiments done with CNRM-CM6-1D, for each experiment, the table indicates the components involved and a short description.**





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




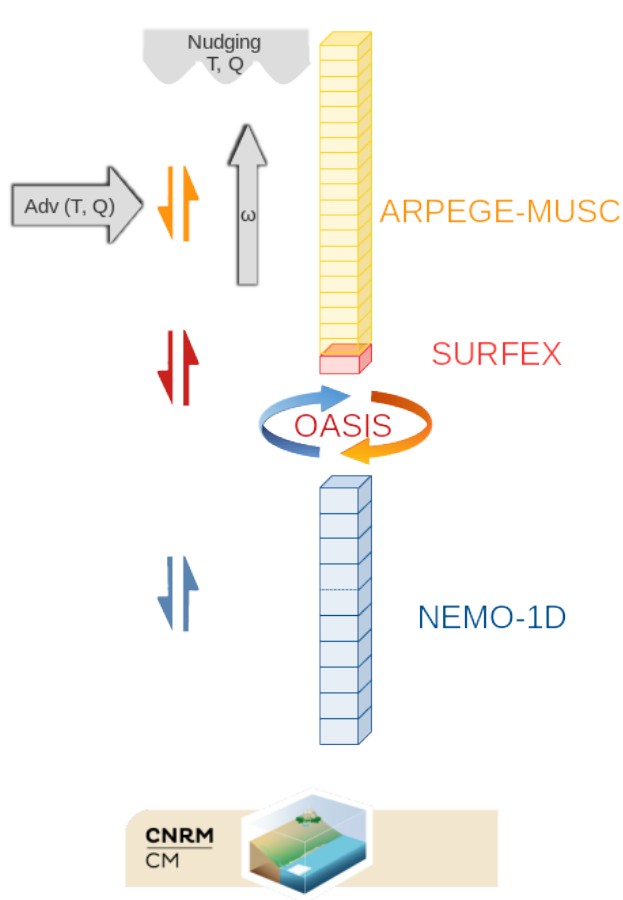

**Figure 1.** Schematic of the CNRM-CM6-1D unicolumn coupled model. The grey elements picture imposed large-scale forcings.





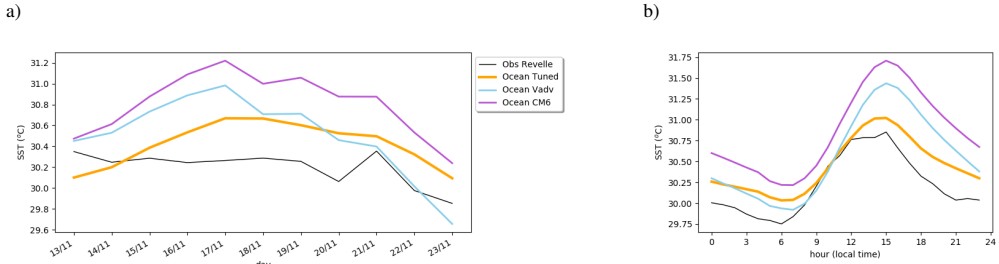

**Figure 2.** a) Daily time-serie and b) mean diurnal cycle of SST ($^\circ C$) in ocean forced experiments compared to SST measured at the R/V Revelle.

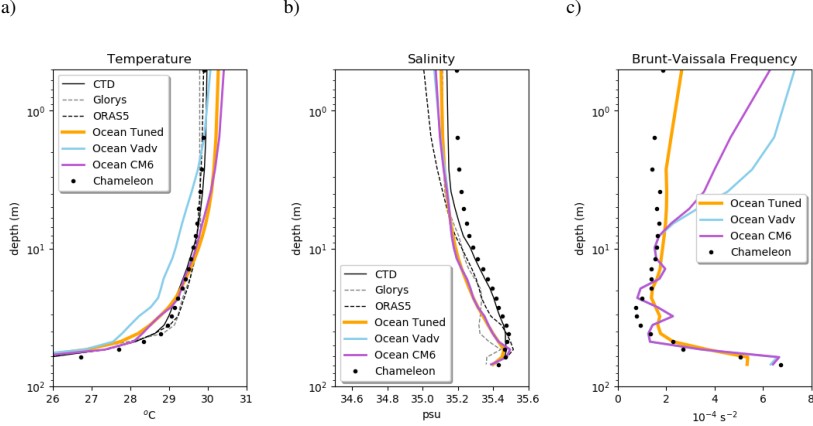

**Figure 3.** Mean vertical profile of a) temperature (in $^\circ C$), b) salinity (in psu) and c) the brunt-Vaisala frequency (in $10^{-4} s^{-2}$) averaged over the simulated period (13 Nov - 22 Nov).

a)

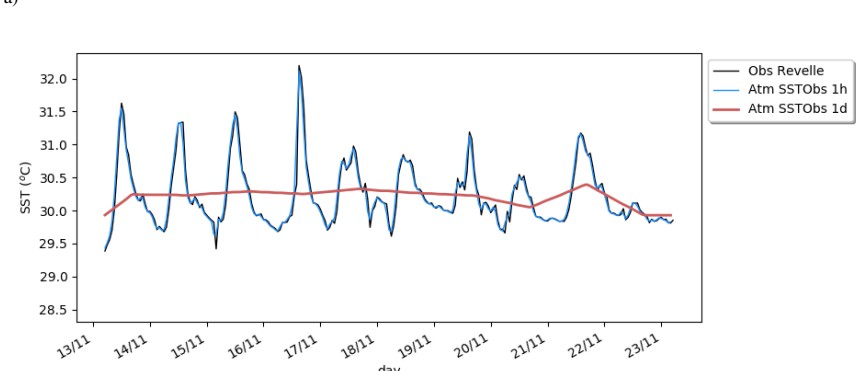

b)

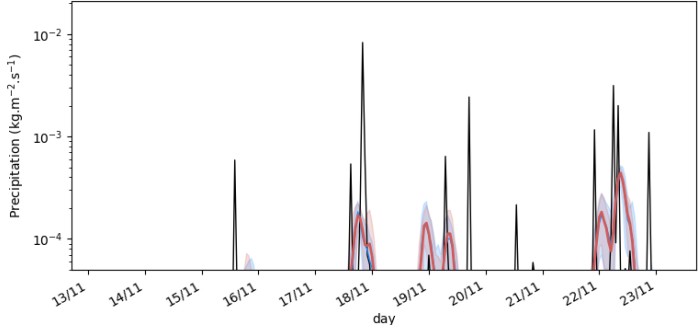

**Figure 4.** Hourly time-serie of a) temperature (in $^\circ C$) and b) precipitation (in kg.m$^{-2}$.s$^{-1}$) in atmospheric forced experiments and observed estimates. The shading represents the spread of the member ensemble given by two standard deviations.



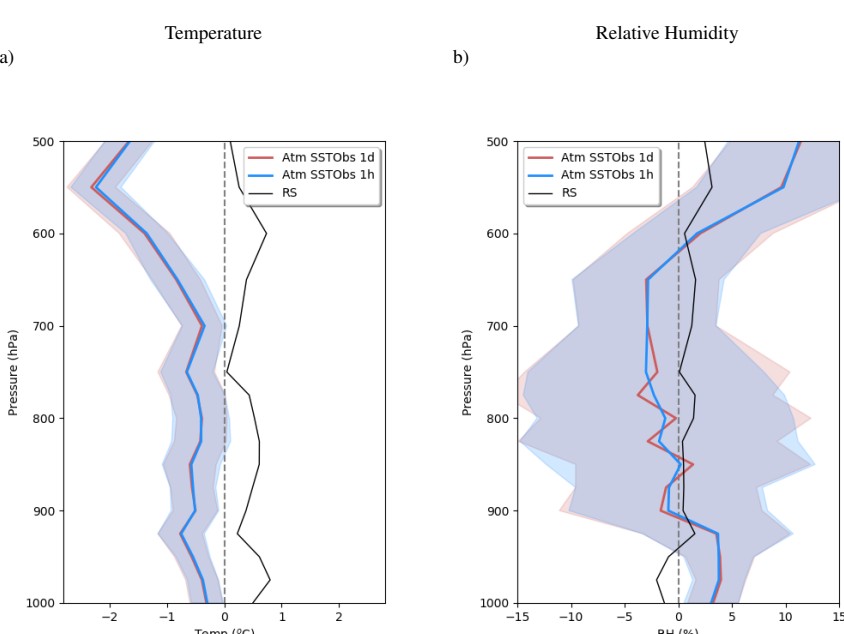

**Figure 5.** Mean difference in the vertical profile of a) temperature (in $^\circ C$) and b) relative humidity (in %) between atmospheric experiments and the Era-Interim reanalysis (Dee et al., 2011) averaged over the simulated period (13 Nov - 22 Nov). The shading represents two standard deviations of the inter-member ensemble. The black line indicates the difference between the local sounding data at R/V Revelle and the ERA-Interim reanalysis over the same period.

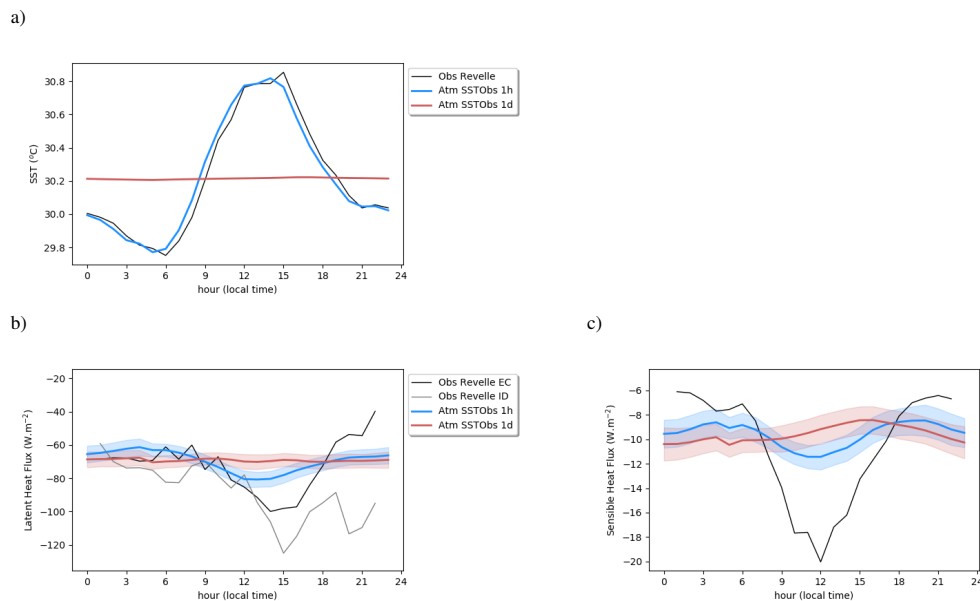

**Figure 6.** Mean daily cycle of a) surface temperature (in $^{\circ}C$), b) surface latent heat flux (in W.m$^{-2}$) and c) surface sensible heat flux (in W.m$^{-2}$) averaged of the simulated period (13 Nov - 22 Nov) for the atmospheric experiments and observed estimates. The shading represents two standard deviation of the inter-member ensemble.

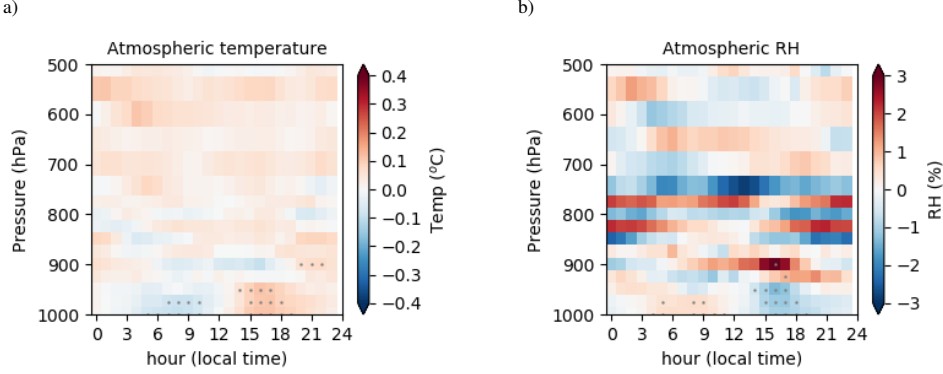

**Figure 7.** Change in mean daily cycle between experiment with hourly SST forcing (Atm SSTObs 1h) and experiment with daily SST forcing (Atm SSTObs 1d) for a) the atmospheric temperature (in $^{\circ}C$) and b) relative humidity (in %). Dots indicate significant anomalies according to a student-T-test at the 95% significance level.

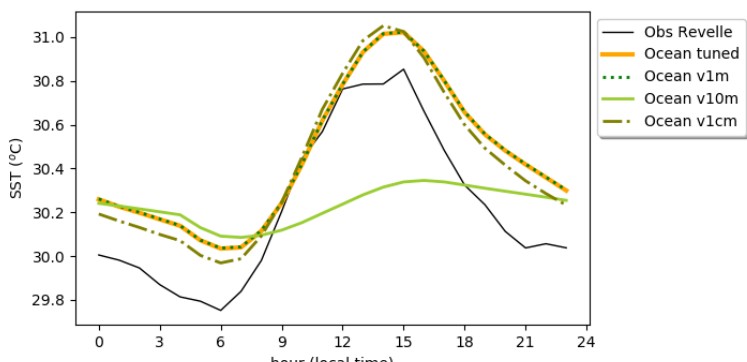

**Figure 8.** Mean daily cycle of surface temperature (in °C) for the reference ocean forced experiment (orange) and sensitivity tests to the ocean vertical discretization (green).

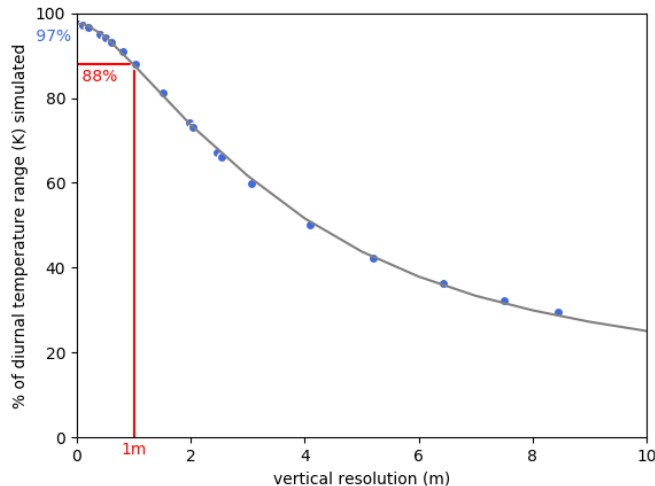

**Figure 9.** Ratio (in %) of the simulated diurnal SST range on the observed range at R/V Revelle, averaged over the first 5 days of the case-study depending on the vertical resolution in the ocean model (blue dots). The grey line represents the ratio estimated from the experiment with the finest vertical resolution (ie 1 $cm$) and by averaging the temperature over the corresponding depth. Note that the statistics could not be computed over the 10 days of the experiments since some vertical resolution experiments where instable after these first 5 days.





a)

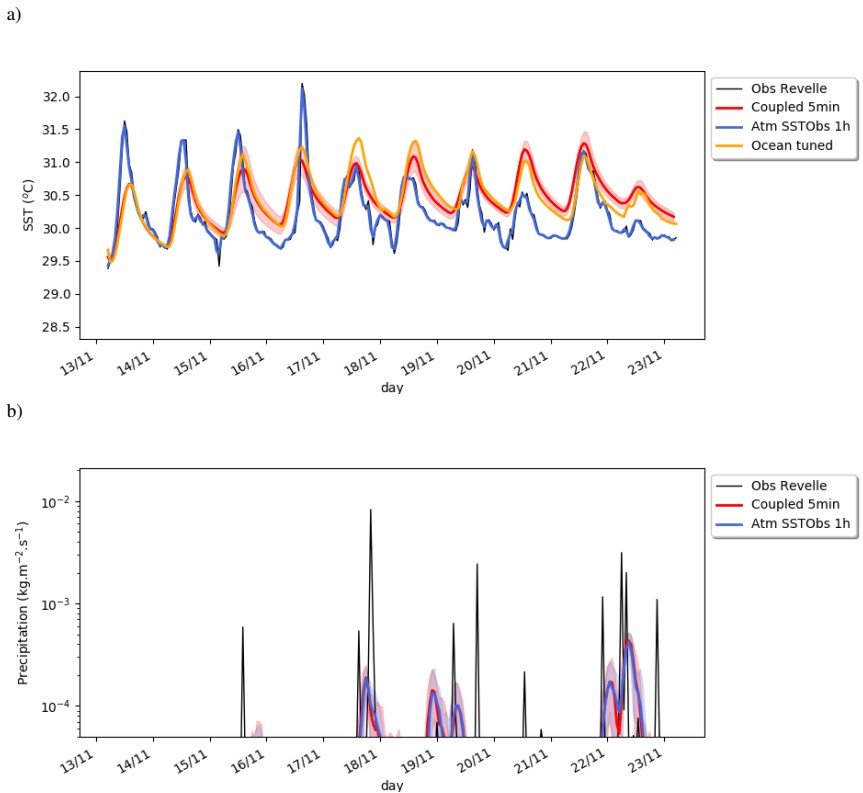

b)

**Figure 10.** Hourly time series of a) surface temperature (in °C), and b) precipitation (in kg.m$^{-2}$.s$^{-1}$) for the coupled experiment (in red) and reference atmospheric (blue) and ocean forced (orange) experiments along with observations at R/V Revelle (black). The shading represents two standard deviations of the inter-member ensemble.



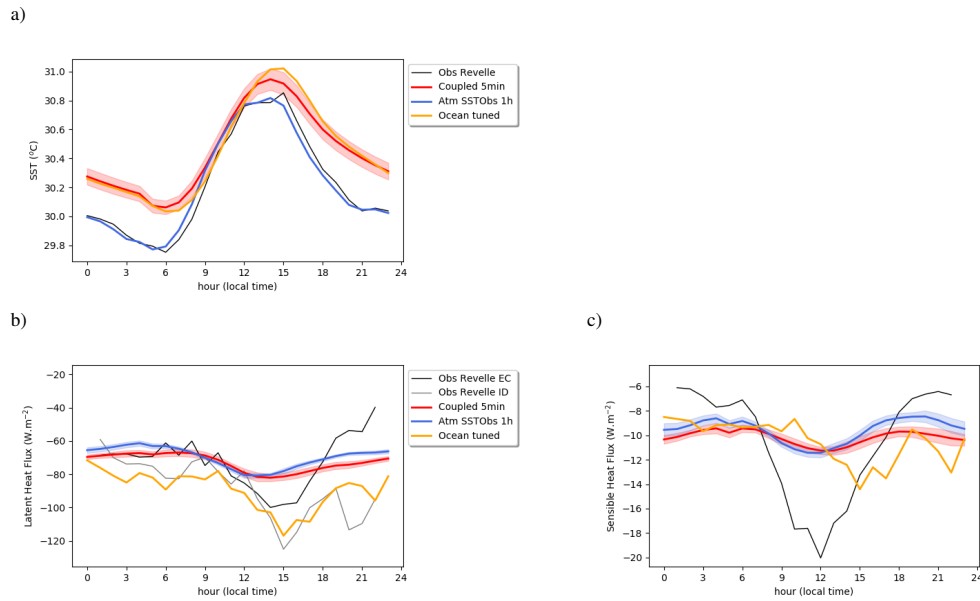

**Figure 11.** Mean daily cycle of a) surface temperature (in °C), b) surface latent heat flux (in W.m$^{-2}$) and c) surface sensible heat flux (in W.m$^{-2}$) averaged of the simulated period (13 Nov - 22 Nov) for the coupled experiment (in red) and reference atmospheric (blue) and ocean forced (orange) experiments along with observations at R/V Revelle (black). The shading represents two standard deviations of the inter-member ensemble.



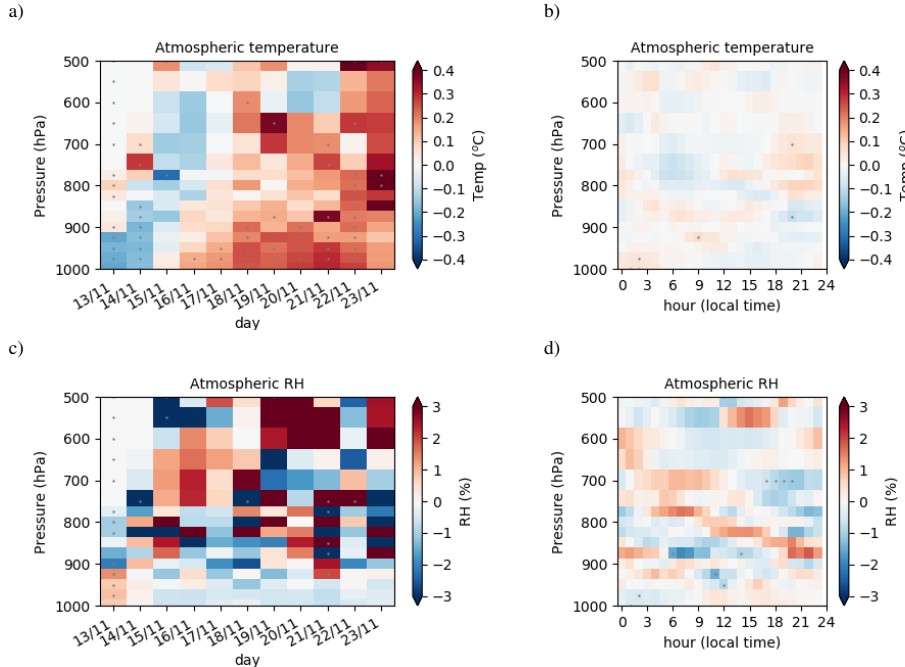

**Figure 12.** Change in mean daily vertical profile of a) temperature (in °C) and c) relative humidity (in %) between reference coupled experiment and the Atm SSTObs 1h experiment. (b,d) represents the respective change in mean diurnal cycle averaged over the 10 days period with the mean bias removed at each level. Dots indicate significant differences according to a student T-test at 95% confidence level.



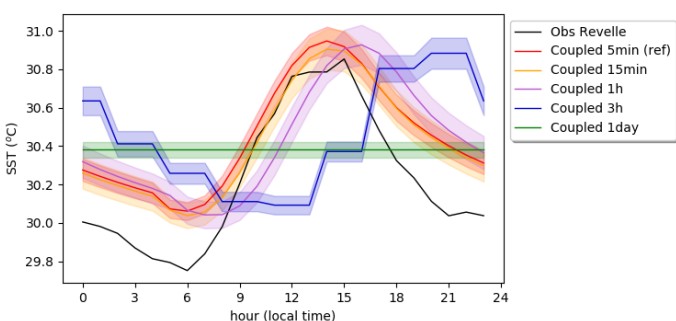

**Figure 13.** Mean daily cycle of SST (in °C) in sensitivity experiments to the coupling frequency averaged over the simulated period (13 Nov - 22 Nov). The shading represents two standard deviations of the inter-member ensemble.

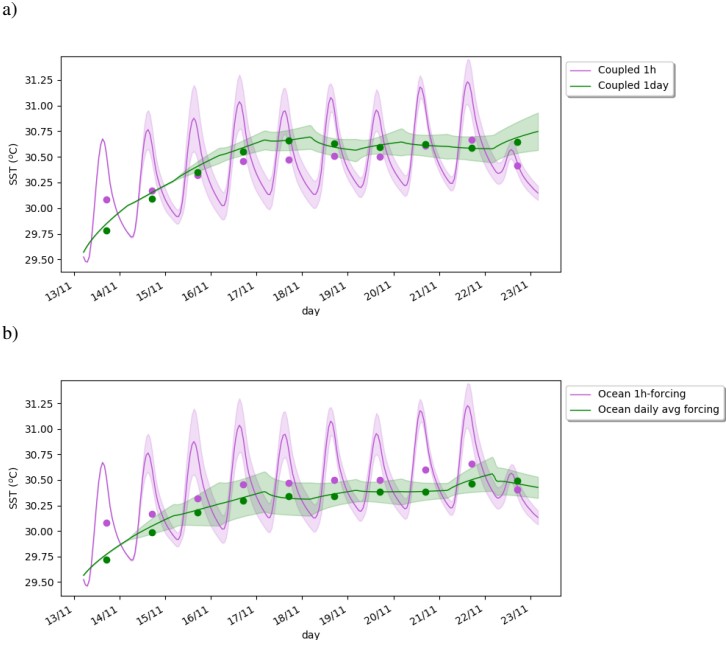

**Figure 14.** Evolution of SST (°C) simulated by the ocean component for a) the hourly and daily coupled experiments, b) ocean forced experiments forced with the hourly coupled atmospheric state taken hourly or as a daily average. The dots indicate the daily mean SST values for each experiments respectively.





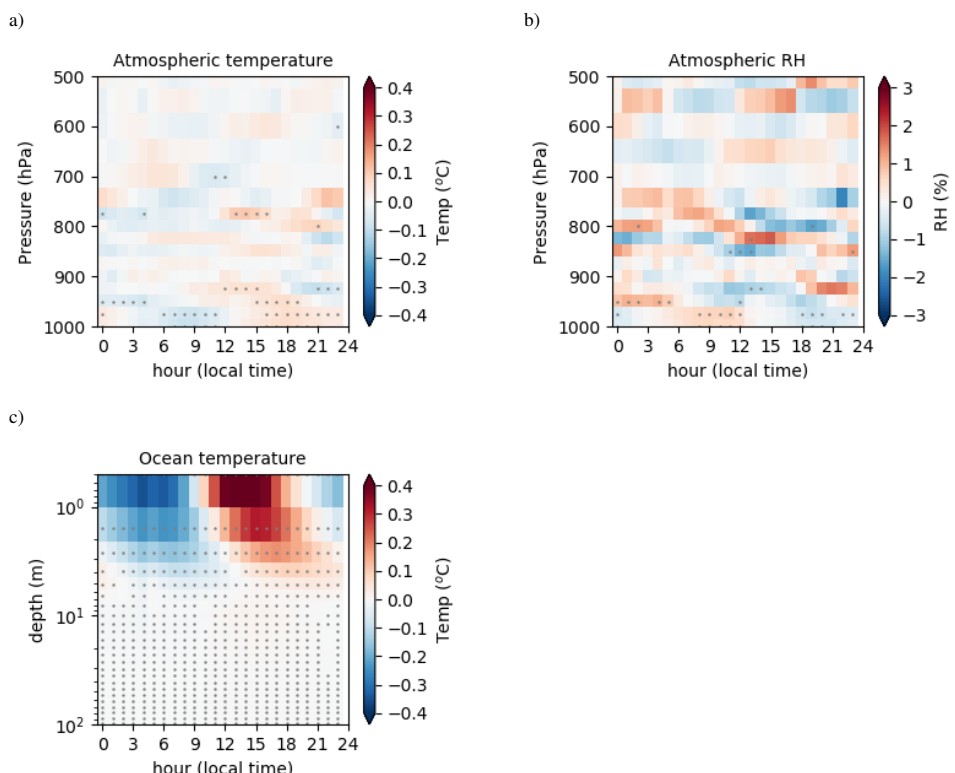

**Figure 15.** Change in mean diurnal cycle with the mean bias removed at each level (right column) between the 5 min coupled experiment and the 1-day coupled experiment for a) the atmospheric temperature (in °C), b) the relative humidity (in %) and c) the ocean temperature (in °C). Dots indicate significant differences according to a student T-test at 95% confidence level.

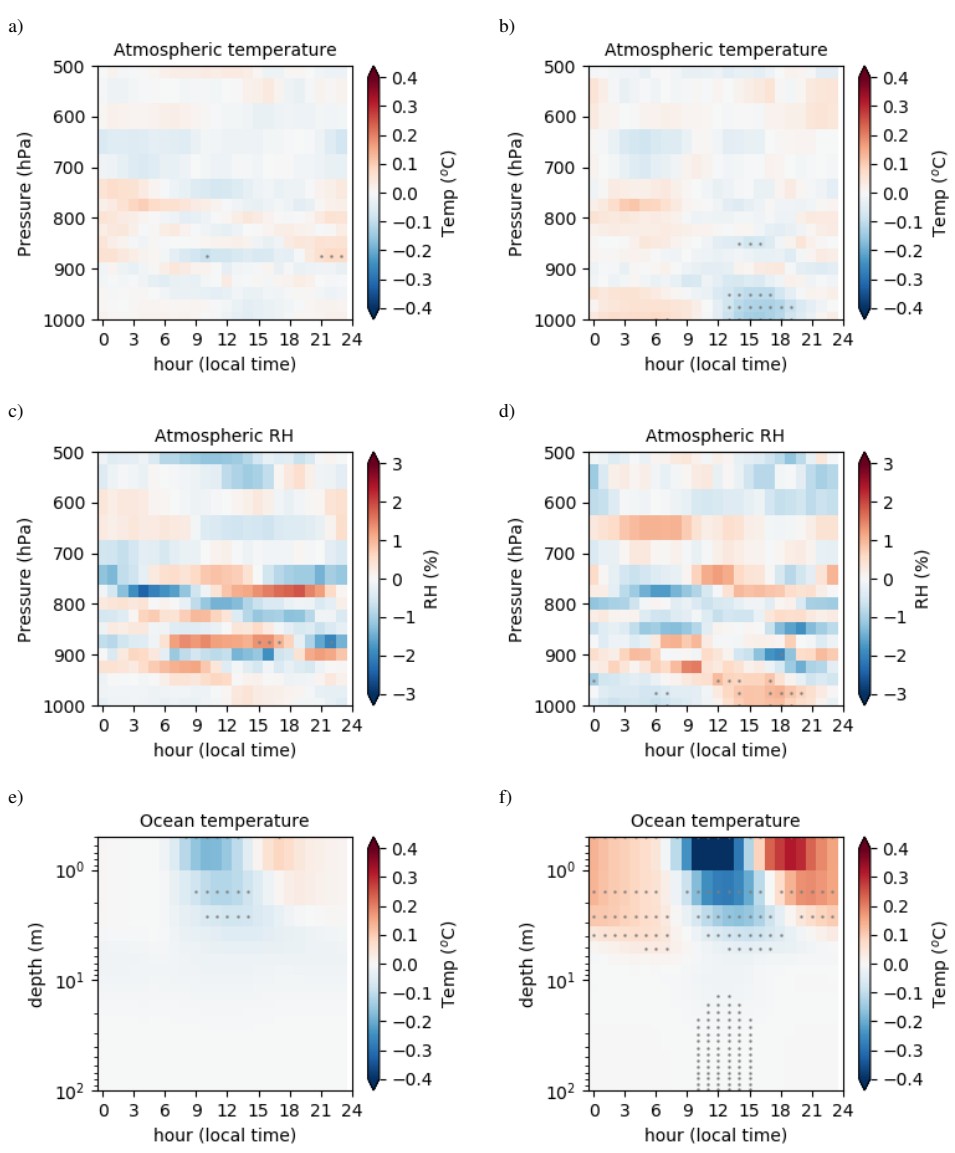

**Figure 16.** Change in mean daily cycle between the 1 h coupling experiment and the 5 min coupled experiment (left) and between 3 h coupling experiment and the 5 min experiment (right) for (a,b) the atmospheric temperature (in °C), (c,d) the relative humidity (in %) and (e,f) the ocean temperature (in °C). Dots indicate significant differences according to a student T-test at 95% confidence level.