# Peer review of "Assessment of the sea surface temperature diurnal cycle in CNRM-CM6-1 based on its 1D coupled configuration"

_Geoscientific Model Development, 2021_

## Author Comment (AC3)

This manuscript presents a newly coupled atmosphere-ocean single column model (AOSCM, CNRM-CM6-1D). It demonstrates the model's ability to simulate the diurnal cycle based on a case study during the Cindy-Dynamo campaign. The authors explore the dependence of skill in modeling the diurnal SST variability on coupling of the components and the coupling frequency. The manuscript is well written, coherent, and presents a relevant scientific contribution. It demonstrates the usability of the new AOSCM and points out several questions that can be investigated with it. I recommend acceptance upon a few minor edits. I list my comments below.

We acknowledge the anonymous reviewer for his/her very positive feedback. We reply to the comments in blue below.

L17 "This suggests that.." This sentence is not clear, please explain.

This sentence has been rephrased to clarify the idea.

L26/27 "either between parameterizations" -- "between parameterized processes" ? It might help the reader if you gave an example.

We have added a reference to Bhattacharya et al. (2018) which illustrated well the feedbacks in between parameterizations in an atmospheric model.

L47 suggest "as is the case in the the real.."

Done

Section 3.3 and 3.4 (and or Table 1) should mention nominal vertical resolutions and active parameterization schemes (i.e. KPP or TKE or ... in the ocean? schemes in the atmosphere?), and nudging / restoration time scales

We have added these pieces of information on lines 163-165 for the atmosphere and lines 196-200 for the ocean. Note that the restoring time-scale of the atmospheric forcing is already given on line 179.

L237 and 238 suggest removing "clearly"

Done

L296 suggest removing "It mean that"

Done

Table 2 / experiment Vadv: is the 0.1degree C cooling throughout the column? Across a level? Across base of the mixed layer?

We have applied a cooling term of 0.1°C throughout the column. We agree that it is not realistic but as the results were similar whatever the chosen vertical extent for this term, we decided to show the test using the most simple choice. We have added this clarification in table 2 and in the main text on line 232. Note also that this term is a cooling term whose origin is not stated (we cannot say whether it represents a vertical advection term or a horizontal advection). We have thus removed the term "vertical" in the table.

Fig 1 rotate epsilon in upward arrow

It is not an epsilon but an omega (the lagrangian tendency or air pressure, ie the vertical velocity in pressure coordinates). This has been clarified in the figure's caption and in the main text.

Fig 5 Why are the profiles shown in reference to ERA-Interim, and not as they are next to each other? Why not in reference to the R/V Revelle soundings that should be more accurate?

It is difficult to say that the observed local profile is more accurate than the ERA-Interim profile in this case. The atmospheric forcing method is representative of an area spanning a 50-km-radius disk centered around the R/V Revelle, while the soundings provide local measurements. Therefore, the comparison with local soundings would be unfair too. We agree that we could have used the sounding profiles as a reference for this figure, but given the large model ensemble spread, our conclusions won't be altered. Therefore, we did not modify the figure.

Fig 15c (and 16f lower part): is the significance correctly indicated here?

Yes, we have checked this. The significance is based on a student t-test which compares the difference in mean values to the standard deviations. As evidenced in figure R1, there is a footprint of the difference in the daily cycle phasing down to 100m, even if the difference is very weak.

[Figure]

*Figure R1: Change in mean daily cycle of ocean temperature at 97m depth between experiments with 1h and 3h coupling time step relative to a 5min coupling time-step.*

Bhattacharya, R., Bordoni, S., Suselj, K., and Teixeira, J.: Parameterization Interactions in Global Aquaplanet Simulations, J. Adv. Model. Earth Syst., 10, 403–420, https://doi.org/10.1002/2017MS000991, 2018.

---

## Author Comment (AC4)

This paper describes the developed of a coupled single column model, and provides an example of it's use to explore the impact of coupling on the representation of the diurnal cycle, and it's sensitivity to certain model choices. Such modelling frameworks are relatively rare and the development of a coupled modelling framework for parametrization development, particular for those parametrizations closely linked to air-sea interaction, would be potentially useful in the development of coupled climate models.

It's not clear that the case (or experiments) chosen or the results shown are the best to demonstrate the potential of this framework, as the results about the sensitivity to coupling timescale or vertical resolution in the ocean can largely be obtained from ocean single column models (as in Bernie et al., 2005). Although, the small sensitivity to coupling is in itself a useful result.

Given the strong role of shortwave radiation in driving the diurnal cycle of SST (and in controlling the strength of it in light wind conditions) I find it surprising that there is no mention of the modelled shortwave radiation or cloud cover particular given the noted differences in precipitation (and hence probably cloud) (see 9&10 below). Overall the analysis of the processes resulting in the small differences between coupled and uncoupled or sensitivity experiments in the coupled framework is limited, and doesn't sell the potential of the modelling framework. The authors miss an opportunity to carry out a more detailed investigation of the impact of the coupling on the rectification of the diurnal cycle, to which this framework is uniquely suited (see 9 below).

We acknowledge the anonymous reviewer for his/her constructive comments. This case study has been developed in the context of a French research project (COCOA) and we have taken this opportunity to develop the single column model in a collaborative environment; it has been used in several studies, in particular Brilouet et al., (2021) assess the effect of oceanic diurnal warm layer in a LES framework. Our conclusion of the small coupling sensitivity is in itself a result worth to be published.

We answer below more specifically to the two main criticisms raised here. Replies are highlighted in blue below.

Specific Comments

1. In lines 156-160 and again in lines 253-255 the authors refer to the fact that the DTR of the mean diurnal cycle and the mean DTR are not the same "as it is a non-linear calculation" and in 254-255 link this to varying amplitudes. This is miss-leading as it is not variations in amplitude which lead these differences but variations in phase (or shape) of the diurnal cycle which can lead to these effects.

We agree that our formulation was not that clear. Our concern was more related to the comparison between simulations and observations. In observations, the SST data is relatively noisy and the quantification of the SST diurnal cycle can be strongly dependent on the occurrence of local events (such as an abrupt change in cloud cover). In fact, the time of the SST maximum depends on the day considered in observations (see figure R1 below). In the model, the simulated SST daily evolution is much smoother and the time of maximum is more similar among days. We agree with the reviewer's comment that it may be due to flaws in the model that may not realistically represent the SST evolution. But, we argue that the comparison is also biased given that the SCM atmospheric forcing is representative of a large area (a 50-km-radius disk centered at the R/V Revelle), while SST observations are

local. Here the type of clouds is typically cumulus clouds. If a cumulus comes above the R/V Revelle, then the input solar radiation will be severely impacted locally. At the model grid scale, cloud cover remains significantly below 1 (cumulus regime) and has a smoother temporal behavior. Nevertheless, the analysis of the 10-day mean diurnal cycle should reduce the mismatch, assuming ergodicity between time and space for the observations.

[Figure]

*Figure R1: SST diurnal evolution for each day of the studied period for the first member of the reference coupled experiment (1h coupling) and the local observed SST.*

We have modified the manuscript text to better justify this choice. See lines 267-268.

2.  In lines 170-172 the description of the forcing of the atmosphere component could perhaps be improved. In particular the figure seems to indicate (and elsewhere in the text – the description of the advection scheme), that the model is given a large-scale vertical velocity which it uses to compute its own vertical advection term, plus an imposed horizontal advection term, and it is also nudged back to the reference profile. It would help to be explicit about some of these choices, particular as in some atmospheric SCM frameworks, the vertical advection is sometime also prescribed, and vertical velocity information is used only when it is required by a parametrization scheme (e.g. in convective triggering).

This has been clarified in the manuscript. See lines 175-178.

3.  Could the authors provide more information on the discussion of the ocean-tuned experiment L227-L234). The opening line of the paragraph suggests that the vertical diffusion is unchanged (and low) during the whole integration, but surely the presence of the a diurnal cycle in the diffusivity is a critical component of the evolution of the diurnal cycle in the ocean (see e.g. figure 5 of Doney et al., Journal of Marine Research, 1995, 341-369, https://doi.org/10.1357/0022240953213133 ). Is this night time cooling dealt with by a non-diffusive mixing term? It would perhaps be helpful here to see a vertical profile of the diurnal evolution of the ocean, e.g an equivalent of figure 15c. It would also help to see the time evolution of the profile over the course of the integration rather than just a mean profile as shown in figure 3.

We have indeed analyzed the night time cooling in more detail when preparing the manuscript. We would have liked to validate in more detail the intensity of the diurnal cycle at depth in the ocean. In the case study, the effect of the diurnal cycle extends to around 10 m in the ocean.

[Figure]

*Figure R2: Evolution of the Brunt-Vaisala frequency in the forced experiments (shown on Fig. 3 in the manuscript) and given by the Chameleon data at 2.6m depth (3rd ocean layer).*

Figure R2, clearly shows that in the "Ocean CM6" and "Ocean Vadv" experiments the Brunt-Vaisala frequency evolution does not depend much on the diurnal cycle. In the "ocean Tuned" experiment, there is a clear diurnal cycle with a near zero value at night. Most likely, the ad hoc "enhanced vertical diffusion" parametrization (Lazar et al., 1999) of ocean convection activates at night as a result of unstable vertical stratification (negative Brunt-Vaisala frequency), which triggers a diurnal cycle of the vertical diffusivity and the associated mixing. This behavior appears as more realistic compared to the Chameleon data, even if we should take this reference with caution.

In our simulations, we have not diagnosed the trend due to the diffusion itself. We agree that it appears to be a nice case study to go a step further on this point and we will pursue this effort in our future work. Besides, figure R2 clearly exhibits that the ocean tuned configuration largely improves the modeled diurnal cycle.

4. In the atmosphere only experiments why is there a phase shift between the RV Revelle SST and the Atmosphere 1hr SST?

We thank the anonymous reviewer for spotting this error. We have used the hourly averages as input data to the model that were used as instantaneous values at the beginning of the average period. We have rerun the simulations by imposing the hourly value at the middle of the average period. It would have been preferable to impose the mean value at each time step but it was not feasible in the model due to technical issues. This results in suppressing the time-lag seen on figure 6a. Note that the model does not reproduce exactly the observed mean cycle due to the online interpolation made between the prescribed values.

All other figures showing "Atm SSTObs 1h" experiment have been updated with this new simulation, and this does not impact any of our results.

5. L267-8 Although the peak amplitude of the precipitation is larger in observations (what horizontal scale and temporal resolution are these), the model precipitation appears to have a longer duration, how to the daily averages compare, although the logarithmic scale would suggest that the extended duration is not enough to account

for the differences. If there is a significant difference in the precipitation (and cloud cover) between the observations and the model even when forced with diurnally varying SSTs the subsequent limited sensitivity to coupling is not surprising.

The model precipitation and the local observations are not directly comparable due to the different spatial scale. We have thus added as reference the precipitation estimated from the radar measurement, which are used as a constraint in preparing the atmospheric forcing. These are therefore consistent with the atmospheric forcing used to drive the SCM. Figures 4 and 10 have been updated accordingly. This shows that the model behaves very realistically, in particular on the 22/11, the amount and duration of precipitation match well with the radar estimates. When comparing the atmospheric forced simulations with and without the diurnally varying SST, we do not observe a systematic change in the precipitation as pictured on figure R3.

[Figure]

*Figure R3: Mean daily cycle of precipitation (kg.m$^{-2}$.s$^{-1}$) averaged of the simulated period (13 Nov - 22 Nov) for the atmospheric experiments forced with daily mean SSTs (Atm SSTObs 1d) and with diurnally varying SSTs (Atm SSTObs 1h). The shading represents two standard deviation of the inter-member ensemble.*

6. L290 Is the daytime drying when forced by hourly SST cf daily SST just a relative humidity effect of changes in temperature or does it reflect a drying in absolute terms. Daytime increases in SST would tend to imply increased LH flux and hence moistening during the day, but increased low-level instability could deepen the atmospheric boundary layer, or promote shallow convection that could mix humidity out of the boundary layer, the relative moistening and cooling at 900hPa in figure 7 could be evidence of this.

Figure R4 shows the mean diurnal cycle of the specific humidity difference between the hourly and daily SST ASCM simulations. The relative humidity differences between the two simulations thus combine both differences in specific humidity and temperature. In particular, the drying between 15h and 18h LT below approximately 925 results from a decrease in specific humidity and an increase in temperature, while just above, the weakly significant moistening results from an increase in specific humidity and a slight cooling. Figure R5 further illustrates the shape of the boundary layer in the two simulations, at 17h LT. In the hourly SST simulation, the boundary layer is slightly more active (increase in TKE). It is also slightly deeper, explaining the drier/moister vertical dipole (both in terms of specific and

relative humidity). There is a slight effect on the altitude at which cloud cover peaks, but it remains weakly significant due to the large spread in the simulation ensemble. This is consistent with higher SST in the afternoon in this simulation. Though consistent, the signal remains weakly significant. We added a few words on this feature in the manuscript (Lines 310-312).

[Figure]

*Figure R4: Same as figure 7b of the manuscript for specific humidity.*

[Figure]

[Figure]

*Figure R5: Vertical profile of a) TKE b) theta, c) Relative humidity and d) specific humidity e) cloud cover in the bottom atmospheric layer at 17h00 (averaged over members and days), the shading represents 2 standard deviations of the inter-member ensemble.*

7. L326-7 I'm not sure of the intended meaning of this sentence "This also tends to show that improving the representation of the nighttime cooling necessitates to better represent the processes involved (Moulin et al., 2018)." Do the authors mean that there is a still a need for improvement of the representation of the nighttime cooling?

As commented on point 3, the representation of the diurnal cycle at depth remains to be validated using appropriate observational data and dedicated model diagnostics. Few words have been added on this point on lines 351-353.

8. L344-350. Whilst the authors are correct to observe that the model DTR is relatively unchanged during the period, they focus on the relatively low DTR period of the second half of the simulations. From figure 10 it would seem that the DTR is comparable to observations during that period (with the possible exception of 20/11), when it is perhaps a bit large and 19/11 when it may be a bit small. However, during the early period it seems that the model seriously underestimates the DTR, and that it is perhaps during this period that the model is most at fault. Figure 10 also seems to suggest a potential weakness of the authors method of measuring the diurnal cycle in that the model seems to have a relatively regular diurnal cycle (in terms of it's phase and shape) whereas the observed diurnal cycle shows more variability in shape and possibly timing. This may well be a reflection of the impact of individual clouds passing overhead, where as the model sees only the grid-box mean cloud field.

We agree that the model behavior deserves a more detailed analysis. The underestimation of the DTR during the first days is similar in coupled and ocean forced simulation. As in ocean forced mode, the atmospheric forcing is derived from local observations, this supports the idea that biases are not due to the atmospheric forcing but rather to the ocean model itself. Even if the representation of DTR vertical profile could not be validated in detail (see

answer to question 3), the model is probably too diffusive to properly represent such large DTRs. A comment has been added in the manuscript on lines 370-376.

9. L410 on the rectification effect in the coupled model, this is a rather interesting result and a nice example of the potential value of this framework, but the authors do not give it the attention it deserves given the purpose of the paper. Why does the rectification effect not occur in the coupled framework? Is it because the non-linearity in the latent heat flux and upwelling LW radiation acts as a net cooling effect, or are there changes in the SW flux introduced by the coupling which don't appear in the forced simulation, which offset the rectification when both models have the same forcing?

We agree with the reviewer that it is a very valuable point of the study, which deserves a better emphasis. We hopefully better discuss this point in the revised version (lines 441-449), and provide hereafter further discussion. When performing the study in ocean forced mode, the energy provided to the ocean is the same whenever the diurnal cycle is taken into account or not by construction. In coupled mode, we know that it is not ensured but the daily and the hourly coupled experiments are intrinsically different because of the asynchronous coupling. Due to the asynchronous coupling, the ocean receives the atmospheric fluxes calculated over the former coupled period, i.e. from the former day in case of daily coupling (see figure R6 for an illustration with the total net non-solar heat flux). This means that the SST evolution follows the former day cloud radiative forcing. Therefore the rapid equilibrium between atmosphere and ocean is lost. On the contrary, with a 1-hour coupling frequency, the effect of the diurnal SST variations more rapidly feeds back on the atmosphere during the day despite the 1-hour delay. From figure R6, we can see that the net non solar heat flux calculated in the atmospheric component is not strongly modified when changing the coupling period. On the contrary the daily net non solar heat flux received on the ocean side is largely impacted on a daily basis. The ocean receives the flux one day later. Therefore, when comparing coupled experiments, there is not only the effect of the diurnal cycle but also the effect of the coupling delay that is more pronounced. To go a step further, we would need to do additional experiments, probably with a different time-coupling algorithm which is beyond the scope of this study.

[Figure]

Figure R6: surface net non solar heat flux (long-wave + turbulent heat fluxes) as calculated by a) the atmospheric component, and b) as seen in the ocean component for the coupled simulations at 1h and daily coupling time-step. The dots indicate the daily means.

10. L420-425 The discussion of the impact of the coupling frequency on the diurnal cycle might be better illustrated by showing e.g. line graphs of the temperature (or RH) at a given height (or depth) in the respective boundary layers at simply noting that the effect is limited to the lowest (shallowest) height (depth) as in previous figures

Please find below examples of figures drawn from your advice (Fig. R7). We think that these figures are not better than the original one and as we lose information on the vertical extent of the impacts, we have kept the original figure 16.

*Figure R7: change in mean daily cycle 3h-5min, 1h-5min, at 950hPa for atmospheric temperature (a) relative humidity (b) and ocean temperature at 10m*

11. L426-7 The authors indicate there is little impact on the diurnal cycle of cloud cover when changing coupling frequency, but do not indicate anywhere, what the that diurnal cycle of cloud cover is. Indeed, there is no mention of clouds anywhere else in the document, which I find surprising given that one might expect the diurnal cycle to impact the clouds cover (e.g. Ruppert and Johnson, JAMES, 2016, https://doi.org/10.1002/2015MS000610), but make no mention of the impact of the presence or absence of the diurnal cycle on the clouds.

We agree that this point deserves some explanations. Figure R8 below shows the cloud radiative effect at the surface, as simulated in atmospheric forced experiments. Obviously, this effect has a very large inter-member spread every day, except for the first and last days of the simulation. It suggests that the cloud amount and the associated cloud radiative effects are weakly constrained by the large-scale forcing. The use of only 10 days might also limit our ability to extract a robust signal. In the sensitivity experiments to the SST diurnal cycle, the mean surface cloud radiative effect and its spread are not much impacted by the change. Given the large spread and the small impact, we had chosen to discard this aspect in the original manuscript but we agree that this information is important in itself and should be included. This figure has been added as a new panel on figure 4 and comments added on lines 283-286 and on lines 465-466.

[Figure]

*Figure R8: Same as figure 4 for the surface cloud radiative forcing in W.m$^{-2}$*

12. L484-5 "This is probably the absence of dynamical feedbacks in 1D configurations": I'm not sure this statement is justified, haven't the authors shown here that the rectification is a result of a lack of thermodynamic coupling between the atmosphere and ocean.,

Here "dynamical feedbacks" was not the correct term, we were meaning large-scale dynamics. The sentence has been rephrased accordingly.

Technical comments

There are a number of grammatical errors in the text, I've not picked them all out, but have tried to identity ones that I found made the paper difficult to read or risk a mis-understanding of the meaning.

L202 "The mean salinity is accurate to a depth of a few meters"  might be a better way to

phrase this

Done

L224 "A test in which the initial current profile was imposed through the simulation did not strongly impact the thermal profile nor the upper ocean stability" might be a better way to phrase this.

Done

Fig 9. What's the significance of the blue 97%

This is the value obtained with the highest resolution tested. This has been added in the figure caption.

L363. "This shows the impact .." would be better than "This pictures the impact…."

Done

L407 & L418 suggest replacing "picture" with "show"

Done